# Reconsideration of Temperature Determined by the Excited-State Population Distribution of Hydrogen Atoms Based on Tsallis Entropy and Its Statistics in Hydrogen Plasma in Non-Equilibrium State

**DOI:** 10.3390/e25101400

**Published:** 2023-09-29

**Authors:** Koji Kikuchi, Hiroshi Akatsuka

**Affiliations:** 1Department of Electrical and Electronic Engineering, Tokyo Institute of Technology, 2-12-1-N1-10, O-Okayama, Meguro-ku, Tokyo 152-8550, Japan; 2Laboratory for Zero-Carbon Energy, Institute of Innovative Research, Tokyo Institute of Technology, 2-12-1-N1-10, O-Okayama, Meguro-ku, Tokyo 152-8550, Japan

**Keywords:** non-equilibrium plasma, excitation distribution, excitation temperature, electron temperature, Tsallis entropy, statistical physics

## Abstract

In non-equilibrium plasmas, the temperature cannot be uniquely determined unless the energy-distribution function is approximated as a Maxwell–Boltzmann distribution. To overcome this problem, we applied Tsallis statistics to determine the temperature with respect to the excited-state populations in non-equilibrium state hydrogen plasma, which enables the description of its entropy that obeys *q*-exponential population distribution in the non-equilibrium state. However, it is quite difficult to apply the *q*-exponential distribution because it is a self-consistent function that cannot be solved analytically. In this study, a self-consistent iterative scheme was adopted to calculate *q*-exponential distribution using the similar algorithm of the Hartree–Fock method. Results show that the excited-state population distribution based on Tsallis statistics well captures the non-equilibrium characteristics in the high-energy region, which is far from the equilibrium-Boltzmann distribution. The temperature was calculated using the partial derivative of entropy with respect to the mean energy based on Tsallis statistics and using the coefficient of *q*-exponential distribution. An analytical expression was derived and compared with Boltzmann statistics, and the distribution was discussed from the viewpoint of statistical physics.

## 1. Introduction

In discharge plasmas, excited-state distribution influences various elementary processes. It determines the characteristics of the physical and chemical processes in the plasma. Among the diagnostic techniques, optical emission spectroscopy (OES) has been widely applied to determine temperatures, for example, electron, excitation, vibration, and rotation temperatures from the spectrum [1].

Generally, any kind of dynamics in a physical system is governed by the entropy maximization principle if the physical system is in the state of equilibrium [2]. However, if such a dynamic system involves a multibody interaction, particularly radiation processes such as plasmas with low-electron temperature, it does not always obey the principle of maximum entropy according to the Gibbs entropy as the ideal state in common statistical mechanics [3]. Furthermore, there are many publications up to the present time where it is described that the strong non-equilibrium throughout the whole system is found in several systems such as low-temperature reduced-pressure plasma [4], high-temperature fusion plasma [5], and quark–gluon plasma [6]. When the system has such non-equilibrium characteristics, energy distribution functions do not necessarily follow the Maxwell–Boltzmann distribution. Thus, to uniquely determine the temperature of the system, its distribution function must be approximated to the Boltzmann distribution
(1)$$p_{i}\propto \exp\left(-\frac{E_{i}}{kT}\right),$$
where $p_{i}$ is the probability for the system to be in the *i*-th state, $E_{i}$ is the energy of the *i*-th level, *k* is the Boltzmann constant, and *T* is the approximate temperature. When the distribution function $p_{i}$ is plotted semi-logarithmically against the energy $E_{i}$, which is referred to as a Boltzmann plot, a straight line is obtained for the equilibrium state. However, for the non-equilibrium state, the Boltzmann plot does not follow a straight line, and some approximation is inevitably required.

Based on the Lagrange multiplier method, this probability function $p_{i}$ described with Equation (Equation 1) can be derived by maximizing the Gibbs entropy $S_{\mathrm{Gibbs}}$ in statistical mechanics, which is defined as follows:(2)$$S_{\mathrm{Gibbs}}=-k\sum_{i=1}^{N}p_{i}\ln p_{i},$$
where *N* is the maximum energy level. On the other hand, the probabilistic distribution of the quantum state also derives the Boltzmann factor based on thermal physics. Consequently, the temperature of the equilibrium system $T_{\mathrm{Gibbs}}$ can be derived from the entropy *S* based on statistical mechanics as follows:(3)$$T_{\mathrm{Gibbs}}={\left(\frac{\partial S}{\partial U}\right)}^{-1},$$
where *U* is the internal energy of the system, which can be calculated as the mean energy for discretized energy states such as quantum levels as follows:(4)$$U=\sum_{i=1}^{N}p_{i}E_{i},$$
where $p_{i}$ is subject to constraint under the Boltzmann distribution such as Equation (Equation 1). The temperature $T_{\mathrm{Gibbs}}$ calculated using Equation (Equation 3) coincides with the temperature determined from the slope of the Boltzmann plot in the equilibrium state.

However, since in the non-equilibrium state $p_{i}$ cannot be described using the Boltzmann distribution Equation (Equation 1), its temperature cannot be obtained from the Boltzmann plot. As schematically shown in the Boltzmann plot in Figure 1, the distribution function in the non-equilibrium state cannot be expressed as a straight line. Consequently, to determine the temperature of the non-equilibrium system, the distribution function must be approximated to a straight line, which is generally considered to correspond to the local derivative at each energy value in the Boltzmann plot.

The definition of temperature in the non-equilibrium state, which does not obey the Maxwell–Boltzmann distribution, has received a great deal of attention in recent years [7,8]. For example, Álvarez et al. attempted to describe the out-of-equilibrium free-electrons in cold plasmas and applied the developed theory assuming the electron entropy defined based on the Boltzmann H-theorem [7]. Their theory concluded that the partial derivative of the entropy with respect to the electron mean energy as shown in Equation (Equation 3) corresponds to the so-called effective temperature $T_{\mathrm{eff}}$ simply given as
(5)$$T_{\mathrm{eff}}=\frac{2U}{3k}=\frac{2}{3k}\int_{0}^{\infty }\epsilon f\left(\epsilon \right)d\epsilon ,$$
provided that the electron energy distribution function (EEDF) *f* is differentiable and
(6)$$\begin{array}{ccc} \lim_{n\to \infty }v^{3}f\ln f & = & 0, \end{array}$$
(7)$$\begin{array}{ccc} \lim_{n\to \infty }v^{3}f{\left(\frac{\partial S}{\partial n}\right)}_{\epsilon ,n_{e}} & = & 0 \end{array}$$
where *v* is the electron velocity and ϵ is the electron energy.

However, in the realistic conditions of plasma, the relation $T_{\mathrm{eff}}=T_{\mathrm{Gibbs}}$ does not always hold [8], and they solved the Boltzmann equation to obtain the EEDF of oxygen and nitrogen plasmas with inelastic collision cross-sections and calculated the Gibbs entropy. Eventually, these approaches made it possible to define some kinds of temperatures even in non-equilibrium states. However, Equation (Equation 3) derived from Gibbs statistics should not be mathematically appropriate to apply to dynamic systems, because the entropy maximization principle cannot be satisfied for systems in non-equilibrium states.

Therefore, it is worth introducing Tsallis statistics, which has been studied as an improved alternative to the Gibbs statistics [9]. The most fundamental formalism in Tsallis statistics is based on maximizing the Tsallis entropy $S_{q}$ defined as follows:(8)$$S_{q}\equiv -k\frac{\sum_{i}p_{i}^{q}-1}{q-1},$$
The Tsallis entropy agrees with the Gibbs entropy and the one divided by the Boltzmann constant also agrees with the Shannon entropy
(9)$$S=-\sum_{i}p_{i}\ln p_{i}=S_{\mathrm{Gibbs}}/k$$
in the case of the limit as q→1. Tsallis statistics extend the exponential distribution to *q*-exponential distribution, which has a power-law distribution. The intuitional physical meaning of the parameter *q* in Tsallis statistics has been investigated by adapting the sedimentation model that is restricted by the previously sedimented particles, which is a relatively simple stochastic process [10]. Additionally, the interpretation of the nonextensivity parameter *q* is characterized by a fluctuation of the inverse temperature as long-range interactions and long-range microscopic memories [11,12].

Many researchers have investigated non-equilibrium phenomena using Tsallis entropy, for example, in fluxes of cosmic rays [13], Earth’s magnetosphere [14], turbulence [15,16], electron-positron annihilation [17], black hole [18,19], Bose–Einstein condensation [20], dielectric breakdown [21], and plasma physics [22]. In this study, Tsallis statistics is applied to excited-state populations of hydrogen plasma, which is in non-equilibrium state, by a fitting procedure with the *q*-exponential distribution. Based on Tsallis statistics, the temperature can be uniquely determined from the distribution in a strongly non-equilibrium state that follows *q*-exponential-type probability function. However, it is quite difficult to fit the excited-state population distribution with *q*-exponential distribution because such fitting requires a solution to the self-referential equation. Therefore, a self-consistent iterative scheme was adopted, which is similar to the Hartree–Fock method, to calculate *q*-exponential distribution to find the eigen value energy of a quantum many-body system in a stationary state. Finally, the temperature was determined and discussed from the non-equilibrium distribution using Tsallis statistics.

No study has applied Tsallis statistics to determine the temperature relevant to excited-state populations in non-equilibrium plasmas because it is quite difficult to obtain the temperature as a self-consistent solution. In this study, the excitation temperature of non-equilibrium plasma is determined using Tsallis statistics, and the results are compared with those obtained using Boltzmann statistics. As Álvarez et al. studied from the viewpoint of statistical physics, the temperature is given as a reciprocal value of the partial derivative of the entropy with respect to the internal energy [7]. On the other hand, Akatsuka et al. examined another concept of the electron temperature of the non-equilibrium plasma on the relation between the entropy and the energy through the distribution function [8]. If the distribution function is known, the entropy of the system can be calculated, without concepts on equilibrium physics such as “free energy”. Based on the backgrounds described so far, the objective of the present study is to reconsider the “excitation temperature” of the non-equilibrium plasma with non-Boltzmann distribution from the viewpoint of statistical physics by applying the concept of Tsallis entropy.

## 2. Background

### 2.1. Tsallis Statistics

First, it is demonstrated that the temperature derived from Tsallis statistics should follow the *q*-exponential distribution function. Dynamic systems are governed by the entropy maximization principle in equilibrium state. Hence, the state of equilibrium is computed by maximizing the entropy subject to constraints, using a variational function [23]. Boltzmann stated that the occupation probabilities $\left\{p_{i}\right\}$ of the most probable state at equilibrium are those that maximize entropy. Tsallis statistics extended the theory by maximizing the Tsallis entropy, which also satisfies two constraints on the total number of states *W* and on the *q*-average energy $U_{q}$ as follows [24].
(10)$$\sum_{i=1}^{W}p_{i}=1,$$
and
(11)$$U_{q}=\frac{\sum_{i=1}^{W}p_{i}^{q}\epsilon_{i}}{\sum_{j=1}^{W}p_{j}^{q}}.$$
The *q*-average energy is often applied as the extension of mean energy instead of a general average energy, because mean value or dispersion of typical power-law distribution such as Pareto distribution, Lévy distribution, Cauchy distribution, and *t*-distribution, is not finite under the particular conditions. By applying the Lagrange multiplier method, this probability function can be derived as
(12)$$p_{i}=\frac{1}{Z_{q}\left(\beta \right)}\exp_{q}\left[-\frac{\beta }{\sum_{j=1}^{W}p_{j}^{q}}\left(\epsilon_{i}-U_{q}\right)\right]$$
where $Z_{q}\left(\beta \right)$ is the generalized partition function,
(13)$$Z_{q}\left(\beta \right)=\sum_{i=1}^{W}\exp_{q}\left[-\frac{\beta }{\sum_{j=1}^{W}p_{j}^{q}}\left(\epsilon_{i}-U_{q}\right)\right].$$
In the above equation, β is the Lagrange multiplier, and $\exp_{q}$ is the *q*-exponential function defined by
(14)$$\exp_{q}\equiv \left\{\begin{array}{cc} {\left[1+(1-q)x\right]}^{\frac{1}{1-q}} & \mathrm{if}\;1+(1-q)x>0\\ 0 & \mathrm{otherwise}. \end{array}\right.$$
By combining Equation (Equation 12) and other constraints, the following self-referential function
(15)$$p_{i}=\frac{1}{Z_{q}\left(\beta \right)}\exp_{q}\left[-\beta_{q}\left(\epsilon_{i}-U_{q}\right)\right],$$
can be derived, which includes the term probability function $\left\{p_{i}\right\}$ at both sides. The *q*-Gaussian type function can be derived maximizing the Tsallis entropy not only under a first moment constraint but also under a second moment constraint [25]. Now, the *q*-partition function $Z_{q}\left(\beta \right)$ is rewritten as
(16)$$Z_{q}\left(\beta \right)=\sum_{i=1}^{W}\exp_{q}\left[-\beta_{q}\left(\epsilon_{i}-U_{q}\right)\right],$$
where $\beta_{q}$ is defined as
(17)$$\beta_{q}\equiv \frac{q}{q+(1+\alpha )(1-q)}\beta ,$$
and α is the Lagrange multiplier associated with the normalization constraint. Because $\beta_{q}$ is a coefficient of energy, $T_{q}$ is defined as
(18)$$T_{q-\beta }=\frac{1}{k\beta_{q}}.$$
The obtained $T_{q-\beta }$ coincides with the physical temperature derived in accordance with the zero-th law of thermodynamics [26], which is calculated as $T_{q-\mathrm{Tsallis}}$ as follows [24]:(19)$$T_{q-\mathrm{Tsallis}}=\left(1+\frac{1-q}{k}S_{q}\right){\left(\frac{\partial S_{q}}{\partial U_{q}}\right)}^{-1}.$$
Equations (Equation 18) and (Equation 19) hold in cases where the probability function is a function of $\exp_{q}$. The dependence of the *q*-exponential distribution function $p_{i}$ defined in Equation (Equation 15) is shown in Figure 2. The Boltzmann plot becomes curved as the value of the parameter *q* deviates from 1, which indicates some kind of non-equilibrium distribution.

On the other hand, the suprathermal distribution is like κ-distribution as follows:(20)$$B\left(E\right)={\left(1+\frac{\beta_{0}E}{\kappa +1}\right)}^{-\kappa -1},$$
which is a power-law distribution derived from a system with the fluctuations of an intensive quantity as the most fundamental phenomena by maximizing the Gibbs entropy $S_{\mathrm{Gibbs}}$ in superstatistics [12], where $\beta_{0}$ is the average value of the fluctuation of the inverse temperature in superstatistics. Moreover, the parameter κ shapes the suprathermal tail of the distribution and measures its deviation from the Maxwell–Boltzmann equilibrium which is recovered in the limit κ→∞. The κ-distribution is considered to be related to the *q*-distribution in Tsallis statistics assuming (κ+1)=1/(q−1).

### 2.2. Collisional Radiative Model (CR-Model)

The collisional radiative model (CR-model) is a suitable analysis model to describe the excited-state population balance to analyze line-spectroscopic characteristics based on various elementary processes in plasma. It is mainly used to calculate the source terms of balance equations between excited states of discharged atomic species [27]. Studies elucidate that elementary processes of plasmas are mainly governed by electron collision and radiation process, and consequently, the excitation kinetic model is referred to as the “collisional-radiative” model [28,29,30,31,32].

The rate equations of number densities on each excited level *i* follow the population-balance equations, which can be described as
(21)$$\begin{array}{ccc} \frac{dn(i)}{dt}= & - & \left[n_{e}\left(\sum_{j>i}C_{i,j}+\sum_{j<i}F_{i,j}+S_{i}\right)+\sum_{j<i}\Lambda_{i,j}A_{i,j}\right]n\left(i\right)\\ & + & n_{e}\sum_{j<i}\left[C_{j,i}n\left(j\right)\right]+\sum_{j>i}\left[\left(n_{e}F_{j,i}+\Lambda_{j,i}A_{j,i}\right)n\left(j\right)\right]+\left(\alpha_{i}n_{e}+\beta_{i}\right)n_{i}n_{e}, \end{array}$$
which are coupled linear differential equations for any excited levels (i≥2), where $C_{j,i}$ and $F_{i,j}$ are the excitation rate coefficient of electron collision excitation from state *j* to *i* and its inverse de-excitation rate coefficient, respectively, $A_{i,j}$ is the radiative transition probability from state *i* to *j*, $\Lambda_{i,j}$ is its optical escape factor, $S_{i}$ and $\alpha_{i}$ are the electron collision ionization rate coefficient from level *i* and its inverse three-body recombination rate coefficient, respectively, and $\beta_{i}$ is the radiative recombination rate coefficient to level *i*. In Equation (Equation 21), the first line represents the depopulating process from level *i* and the second line represents the populating process to level *i*. The solution of the rate Equation (Equation 21) depends on the electron temperature $T_{e}$, gas temperature $T_{g}$, electron density $n_{e}$, discharge tube radius *R*, and ground-state density n(0). Figure 3 shows a scheme of the elementary processes involved in the CR model of the plasma. When calculating the optical escape factor, the plasma geometry is assumed to be cylindrical with the Doppler line profile [33].

Meanwhile, the relaxation time of the rate equations on the excited states is significantly shorter than that of the mass-average flow, diffusion, or other chemical reaction. This assumption indicates that the plasma is generally in a quasi-steady state in the CR model (Equation 21). The time evolution of number densities on *i*-th level n(i) can be calculated except for the ground state. The number density of the molecular discharge species is determined from its dissociation degree or can be measured by spectroscopic methods such as optical absorption spectroscopy or actinometry method. Therefore, Equation (Equation 21) is approximated to =0 in most practical applications, and consequently, Equation (Equation 21) becomes simultaneous for linear coupled equations for the unknowns n(i)(i=2,3,⋯).

Then, the solution to Equation (Equation 21) is rather easily obtained using a numerical procedure. It is already established that the number density of the excited states in plasmas can be written as a summation of the ionizing term and recombination term. These two terms characterize the excitation kinetics of elementary excited hydrogen atoms in the hydrogen plasma. Hence, the number densities n(i) are deformed as follows:(22)$$n\left(i\right)=n_{0}\left(i\right)+n_{1}\left(i\right),$$
The plasma in which the term $n_{1}\left(i\right)$ is dominant is referred to as an ionizing plasma, while the plasma in which $n_{0}\left(i\right)$ is dominant is referred to as a recombining plasma. When the value of $n_{1}\left(i\right)$ is almost comparable to that of $n_{0}\left(i\right)$, the plasma is referred to as an equilibrium plasma.

## 3. Results and Discussion

### 3.1. The Excited State Population Densities

The number densities n(i) of hydrogen atoms with its principal quantum number i(≥2) in hydrogen discharge plasma are calculated using the CR model Equation (Equation 21), where the population density n(i) can be treated as a function of electron temperature $T_{e}$, density $n_{e}$, and the density of the ground-state hydrogen atoms n(1) in the range of $1\leq T_{e}\;\left[\mathrm{eV}\right]\leq 10$. To calculate the entropy in the present analysis, the population density of the level *i*, n(i) is normalized to deduce the population probability $p_{i}$ with its statistical weight g(i) as follows:(23)$$p_{i}=\frac{n(i)/g(i)}{\sum_{i}\left[n\left(i\right)/g\left(i\right)\right]}.$$
The Boltzmann plot as the CR-model solution to the rate Equation (Equation 21) is shown in Figure 4. The input parameters, except for the electron temperature, are chosen in accordance with the experimental observation in the previous experimental results of OES measurement of microwave discharge hydrogen plasma [34]. Within the parameter range adopted in the present case study illustrated in Figure 4, the simulated plasma was found to be optically thin where the optical escape factor Λ≃1, that is, the effect of optical-absorption is ignorable. It was also confirmed that the relaxation time to reach the thermodynamic equilibrium over the excitation kinetics is much shorter than that of electron–electron Coulomb collisions to establish electron equilibration. Therefore, we conclude that the population distribution of excited-state number densities n(i) is far from the Boltzmann distribution [3,27,35]. The temperature cannot be uniquely determined from the slope of the Boltzmann plot, since the slope in the high-energy region gives a much lower excitation temperature than that in the lower-energy region. These differences in the slope on the Boltzmann plot mean that the excitation temperature depends on the energy region.

On the other hand, in Figure 5, the log-log plot of the excitation population probability $p_{i}$ is shown against the principal quantum number *i*, which is found to be almost a straight line. The discussion on this kind of population kinetics in excited states of hydrogen atoms plasma using the CR model is well conducted in detail [27,34]. The number densities n(i) on the log-log plots on the highly excited region can be described as power-law distribution as $n\left(i\right)/g\left(i\right)\propto i^{-6}$. This is attributed to the fact that the elementary excitation kinetics is dominated by a ladder-like collisional-excitation process. The dominant populating process on the relatively high level that has a power-law distribution is the electron collisional excitation process and the ionizing depopulation process over the electron temperature range from 1 to 10 eV. Especially on excitation level i=39, more than 99% of the population comes from the electron collisional excitation process, while the increase in $T_{e}$ changes the ratio of the dominant depopulation process a little. That is, the ionizing depopulation increases from 88% to 92% and the electron collisional excitation decreases from 12% to 8%. On the other hand, the dominant populating process on the relatively low-level is the electron collisional excitation process or ionization process for the parameter range $T_{e}=1-10$ eV. Especially on excitation level i=2, the increase in $T_{e}$ changes the ratio of the dominant populating process significantly as follows. That is, the electron collisional excitation increases from 60% to 81% for the population. Meanwhile, concerning the dominant depopulating processes, the electron collisional excitation decreases from 72% to 48% while the ionizing process increases from 10% to 51%. Therefore, the change in the dominant populating processes on the relatively low level has a great effect on the *q*-value in the non-equilibrium plasma. The relevance of such power-law distributions to the Tsallis entropy are worth discussing because of the similarity of their *q*-exponential distributions [24,36].

To calculate the temperature of a hydrogen plasma with non-equilibrium properties using the Tsallis entropy, it is necessary to fit the excitation distribution with a *q*-exponential distribution. However, since the *q*-exponential distribution, Equation (Equation 15), is a self-referencing function, it is quite difficult to fit the calculated population distribution to the *q*-exponential distribution; as a result, it cannot be solved analytically. Therefore, as for the algorithm for solving this kind of problem, we applied the self-consistent iterative method in Table 1 similar to the Hartree–Fock method for the determination of the energy and wave function of quantum many-body bodies in the steady state, and eventually, the *q*-exponential distribution was successfully calculated. As a result of the iteration, the $U_{q(k+1)}$ and $U_{q(k)}$ was matched by more than seven significant figures. The population probability $p_{i}$ fitted by the *q*-exponential distribution is shown in Figure 6 and specified in Table 2, together with the value of parameter κ in Equation (Equation 20) in the theory of superstatistics [12]. It is far from the equilibrium Boltzmann distribution and exhibits a curved distribution, especially in the high-energy region. The increase in $T_{e}$ promotes the energy flow like the ionization, which leads to it changing the population processes in plasma, particularly, the electron collisional processes and the ionizing processes enormously. Therefore, the degree of ionization and the population of the elementary processes are reflected to the *q*-value and the κ-value. Figure 6 clearly shows that the fitting using the *q*-exponential distribution captures the non-equilibrium properties calculated using the CR model better than that using the Boltzmann distribution. It is considered that the *q*-exponential distribution function is particularly suitable to describe strongly non-equilibrium distributions. The cumulative value of $\chi^{2}$ is determined almost only by the population density of the level p=2, which is the largest in all the levels, as shown in Figure 6 on a semi-log scale. Meanwhile, the values of $\chi^{2}$ for any excited levels in the case of $T_{e}=2$ eV are less than $10^{-3}$ for the distributions fitted with *q*-exponential function.

### 3.2. Entropy

Generally, the temperature can be determined by the reciprocal of the partial derivative of entropy with respect to mean energy based on Gibbs or Tsallis statistics, as was described in Equation (Equation 3) or Equation (Equation 19), respectively, even though the temperature calculated using Tsallis statistics must be adjusted by *q*-value. Therefore, the slope in the Boltzmann plot, that is, the relation between entropy and mean energy, is discussed. In Figure 7, the relationship between the Gibbs entropy Equation (Equation 2) and the mean energy Equation (Equation 4) is shown in red, while that between the Tsallis entropy Equation (Equation 8) and the *q*-average energy Equation (Equation 11) is shown in blue. As can be seen from this figure, both entropies increase with increasing average energy, and both plots have nearly constant slopes. From this result, we conclude that the Gibbs temperature $T_{\mathrm{Gibbs}}$ as defined in Equation (Equation 3)—the reciprocal of the partial derivative of entropy with respect to the mean energy—does not change much within the range of the plasma parameters set here. However, although the slope in Figure 7 is almost constant for the Tsallis entropy, the Tsallis temperature $T_{q-\mathrm{Tsallis}}$, Equation (Equation 19), should not be considered as simply a constant since the multiplication with the factor $\left[1+\left(1-q\right)\left(S_{q}/k\right)\right]$ must be also conducted in the case of the Tsallis temperature. In addition, from the statistical physics principles, the Tsallis temperature should be determined by the partial derivative adjusted by *q*-value in dynamical systems, which are governed by the principle of Tsallis entropy maximization in non-equilibrium state.

### 3.3. Temperature Determined on Statistical Physics

Various temperatures were calculated and the temperatures calculated using Tsallis statistics were compared with those calculated using Gibbs statistics. Figure 8 shows the comparison between various temperatures defined for the hydrogen plasma, where $T_{\mathrm{Gibbs}}$ and $T_{q-\mathrm{Tsallis}}$ are the temperatures determined from Equations (Equation 3) and (Equation 19), respectively, using the partial derivatives of entropy. Meanwhile, $T_{\langle U\rangle }$ and $T_{\langle U_{q}\rangle }$ is the temperature determined using mean-energy and *q*-mean-energy, respectively. $T_{\langle U\rangle }$ was defined as
(24)$$T_{\langle U\rangle }\equiv \frac{2}{3k}U,$$
where *U* was defined in Equation (Equation 4), whereas $T_{\langle U_{q}\rangle }$ was defined similarly to $T_{\langle U\rangle }$ as follows:(25)$$T_{\langle U_{q}\rangle }\equiv \frac{2}{3k}U_{q}.$$
On the other hand, $T_{q-\beta }$ was defined as Equation (Equation 18), which is, in some sense, a slope of the Boltzmann plot, and in this respect, a kind of extension of the temperature in the Gibbs statistics. Finally, $T_{\mathrm{low}-E}$ is the temperature determined from the differential coefficient of the distribution function at E=10.2 eV, which is expected to be the highest temperature among the temperatures.

Figure 8 shows that each temperature increases monotonically with increasing electron temperature with a similar trend. It should be remarked that $T_{q-\mathrm{Tsallis}}$ was found to be identical to $T_{q-\beta }$, both of which were correctly derived from the Tsallis statistics. Furthermore, it should be noted that $T_{\langle U_{q}\rangle }$ also agrees with $T_{q-\mathrm{Tsallis}}$ and $T_{q-\beta }$, indicating the possibility that the equation
(26)$$U_{q}=\frac{3}{2}kT_{\langle U_{q}\rangle }$$
could be derived from Tsallis statistics as an analogy to Gibbs statistics,
(27)$$E=\frac{3}{2}kT.$$
It must be also remarked that Equation (Equation 26) has not yet been mathematically proved. The relationship that $T_{\mathrm{Gibbs}}$ coincides with $T_{\langle U\rangle }$ also demonstrates the validity of using Gibbs statistics to determine temperature. The *q*-exponential distribution reflects the most probable-state dynamics, where the Tsallis entropy is maximized in the non-equilibrium state.

This study confirmed the relation $T_{\mathrm{low}-E}>T_{q-\mathrm{Tsallis}}>T_{\mathrm{Gibbs}}$. It is reasonable that $T_{\mathrm{low}-E}$ is highest in the temperatures determined from the distribution function, as shown in Figure 8, since any excitation distribution has a larger slope than that of the distribution function at E=10.2 eV. Therefore, the temperature based on Tsallis statistics takes into account the curvilinear distribution of the high-energy bulk region, as opposed to the temperature determined from the local differential coefficient of the distribution function. These excitation temperatures are much lower than the electron temperature $T_{e}$ and close to the gas temperature $T_{g}$.

## 4. Conclusions

Current conventional methods for determining the temperature of non-equilibrium plasmas are incomplete because they ignore the influence of non-equilibrium effects. To overcome this problem by applying the Tsallis statistic, temperatures of the non-equilibrium hydrogen plasma were derived from the distribution function of excited states of hydrogen atoms, which was calculated using the CR model. First, the excited-state density distribution was calculated using the CR model of the hydrogen plasma. The excitation distribution is far from the equilibrium Boltzmann distribution within the present plasma-parameter range which is typically that of the ionizing plasma, because excitation kinetics in the plasma is mainly dominated by a ladder-like collisional process. Therefore, Tsallis statistics was applied to determine the temperature of the excited-state population distribution. To calculate the excitation temperature of hydrogen plasma with non-equilibrium excited-state population distribution, its distribution function must be fitted using *q*-exponential distribution if Tsallis statistics is introduced. However, it was quite difficult to fit the *q*-exponential distribution because it was found to be a self-referenced function that could not be solved analytically. Therefore, a self-consistent iterative scheme was applied to solve the equations using a kind of algorithm similar to that of the Hartree–Fock method.

It was verified that the fitting using the *q*-exponential distribution captures the non-equilibrium properties better than using the Boltzmann distribution. It was found that each temperature defined in the course of this study increased with increasing electron temperature with the same trend. It should be emphasized that $T_{<U_{q}>}$, which is defined as a kind of effective temperature using the *q*-average energy $U_{q}$ as $T_{<U_{q}>}=2\langle U_{q}\rangle /\left(3k\right)$, agrees not only with $T_{q-\mathrm{Tsallis}}=\left[1+\left(1-q\right)\left(S_{q}/k\right)\right]{\left[\left(\partial S_{q}\right)/\left(\partial U_{q}\right)\right]}^{-1}$ but also with $T_{q-\beta }=1/\left(k\beta_{q}\right)$, both of which were defined from Tsallis statistics with the Tsallis entropy $S_{q}$, as an important finding in this study. It was also confirmed using the same procedure that the relation that $T_{\mathrm{Gibbs}}={\left[\left(\partial S\right)/\left(\partial U\right)\right]}^{-1}$ agrees with the effective temperature $T_{\langle U\rangle }=\left(2U\right)/\left(3k\right)$. At the temperatures calculated this time, it was found that there was a relationship of $T_{\mathrm{low}-E}>T_{q-\mathrm{Tsallis}}>T_{\mathrm{Gibbs}}$. Among the temperatures obtained from the distribution function, the highest one was found to be $T_{\mathrm{low}-E}$, which is considered to be a reasonable result.

## Figures and Tables

**Figure 1 entropy-25-01400-f001:**
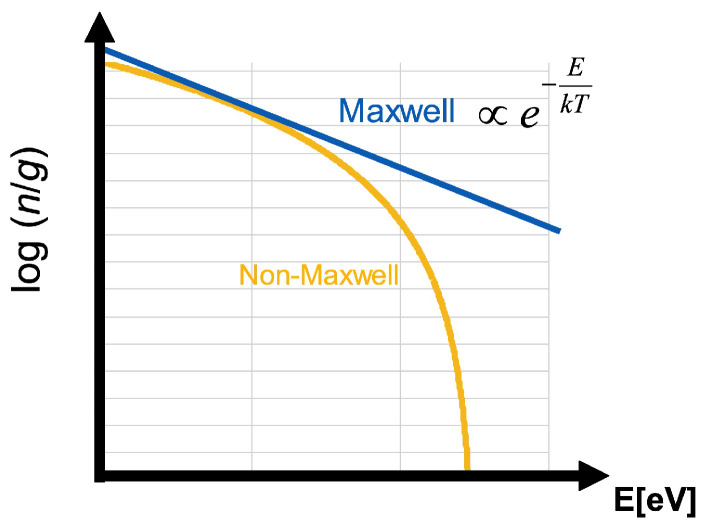
Schematic diagram of Boltzmann plot of Maxwellian and non-Maxwellian distribution functions. On the vertical axis is the logarithm of the number density *n* divided by the statistical weight *g*.

**Figure 2 entropy-25-01400-f002:**
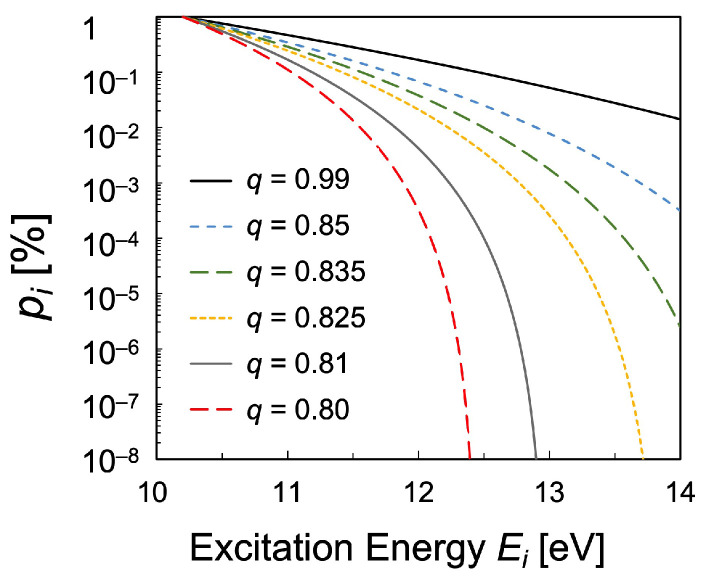
Schematic diagram of the dependence of the *q*-exponential distribution function $p_{i}$ defined in Equation (Equation 15) on the parameter *q*, with constant parameters α=−4,β=1.5 in Equation (Equation 17).

**Figure 3 entropy-25-01400-f003:**
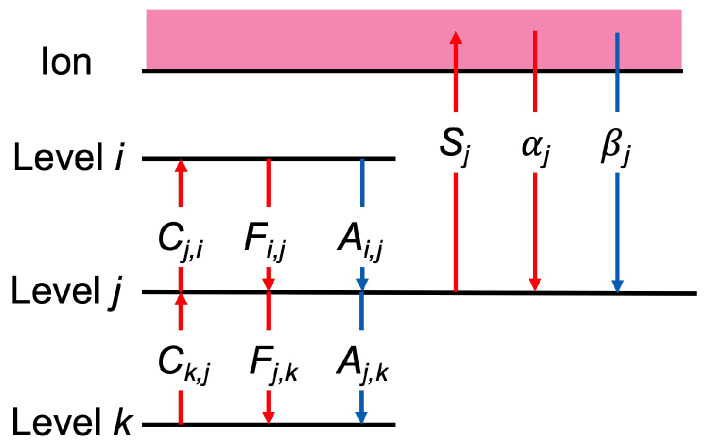
Schematic of the transition of the elementary processes of the level *j* of hydrogen atom in the hydrogen plasma treated in the CR model.

**Figure 4 entropy-25-01400-f004:**
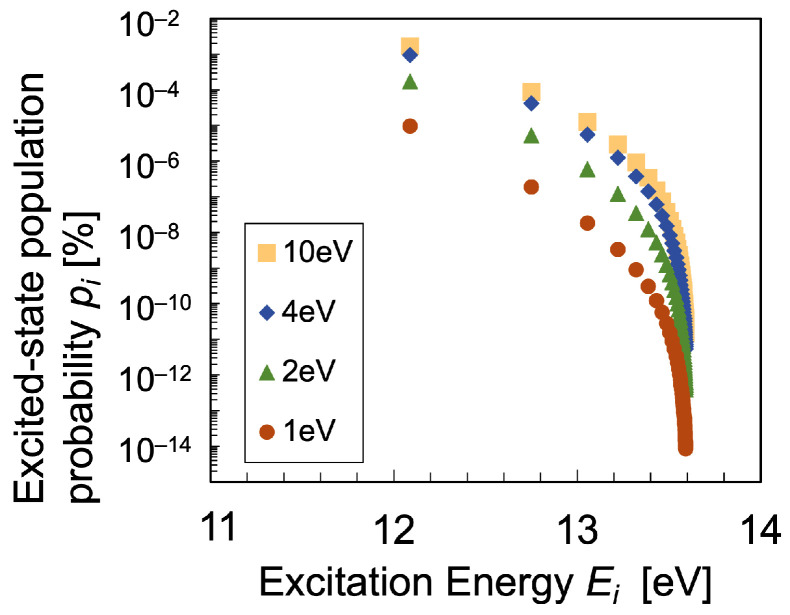
Boltzmann plot of the excited-state population probability $p_{i}$ calculated using the CR model under the condition: gas temperature $T_{g}=0.05\;\mathrm{eV}$, electron density $n_{e}=5\times 10^{13}\;{\mathrm{cm}}^{-3}$, discharge pressure P=1 Torr, and the hydrogen dissociation degree 0.05%, for electron temperature range $1\leq T_{e}\;\left[\mathrm{eV}\right]\leq 10$.

**Figure 5 entropy-25-01400-f005:**
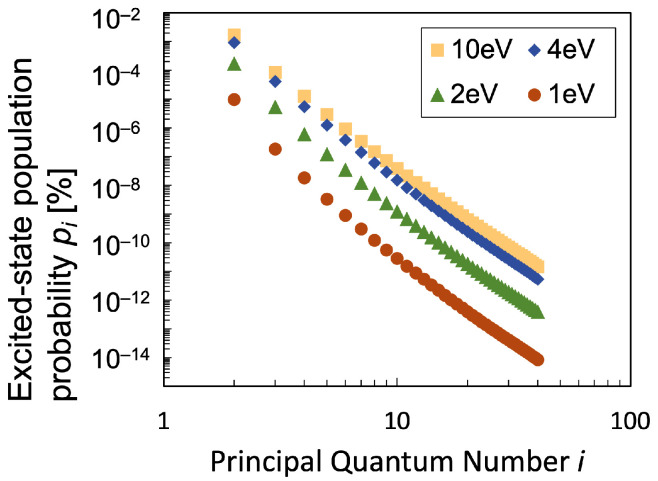
Log-log plot of the excited-state population probability $p_{i}$ calculated using the CR model under the same condition as in Figure 4: gas temperature $T_{g}=0.05\;\mathrm{eV}$, electron density $n_{e}=5\times 10^{13}\;{\mathrm{cm}}^{-3}$, discharge pressure P=1 Torr, and the hydrogen dissociation degree 0.05%, for electron temperature range $1\leq T_{e}\;\left[\mathrm{eV}\right]\leq 10$.

**Figure 6 entropy-25-01400-f006:**
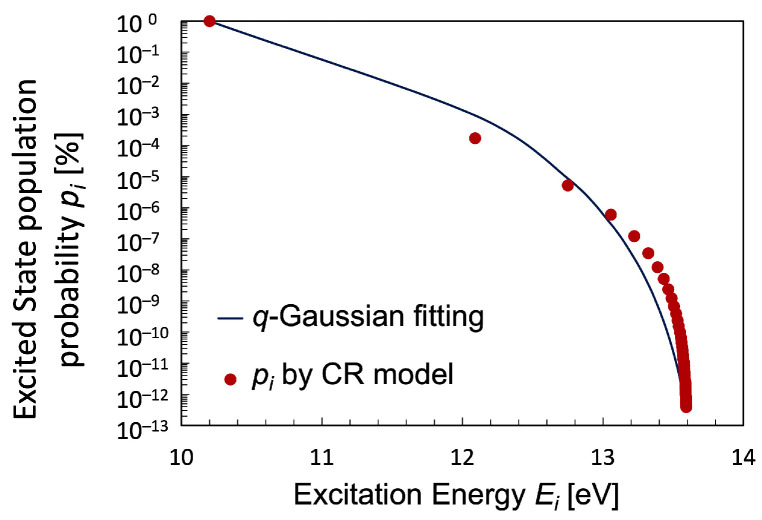
Fitting curve of the Boltzmann plot of population probability function $p_{i}$ of hydrogen atom with *q*-exponential distribution of q=0.89 for the hydrogen plasma with $T_{e}=2$ eV. Other plasma parameters are chosen to be the same as in Figure 4 and Figure 5.

**Figure 7 entropy-25-01400-f007:**
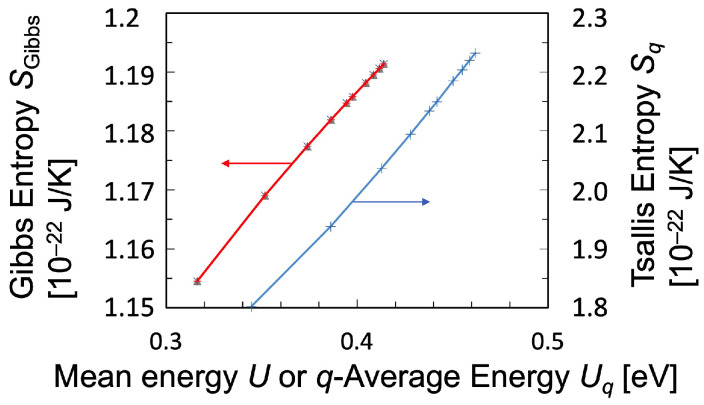
Dependence of entropy plotted against mean energy or *q*-average energy. Red: Gibbs entropy vs. mean energy. Blue: Tsallis entropy vs. *q*-average energy. The plasma parameters are chosen to be the same as in Figure 4. The electron temperature $T_{e}$ in the CR model is scanned within the range from 1 to 10 eV.

**Figure 8 entropy-25-01400-f008:**
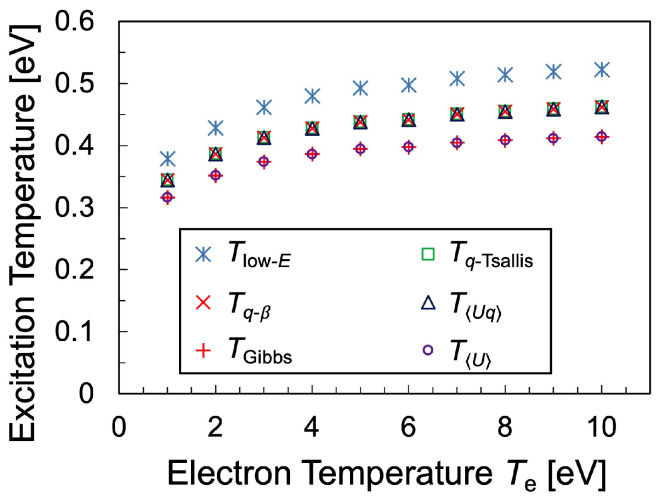
Comparison between various excitation temperatures plotted against electron temperature with the gas temperature $T_{g}=0.55$ eV, electron density $n_{e}=5\times 10^{13}\;{\mathrm{cm}}^{-3}$, discharge pressure 1 Torr and dissociation degree of hydrogen molecule 0.05%.

**Table 1 entropy-25-01400-t001:** Iterative scheme to solve self-referential function.

1: Initialize	
	$Z_{q(0)}\left(\beta \right)=\sum_{i=1}^{W}\exp_{q}\left(-\beta_{q}\epsilon_{i}\right)$
	$p_{i(0)}=\frac{1}{Z_{q(0)}\left(\beta \right)}\exp_{q}\left(-\beta_{q}\epsilon_{i}\right)$
	$U_{q(0)}=\frac{\sum_{i=1}^{W}p_{i(0)}^{q}\epsilon_{i}}{\sum_{j=1}^{W}p_{j(0)}^{q}}$
2: Iterate	k=0,1,2,… until convergence
	$Z_{q(k+1)}\left(\beta \right)=\sum_{i=1}^{W}\exp_{q}\left[-\beta_{q}\left\{\epsilon_{i}-U_{q(k)}\right\}\right]$
	$p_{i(k+1)}=\frac{1}{Z_{q(k+1)}\left(\beta \right)}\exp_{q}\left[-\beta_{q}\left\{\epsilon_{i}-U_{q(k)}\right\}\right]$
	$U_{q(k+1)}=\frac{\sum_{i=1}^{W}p_{i(k+1)}^{q}\epsilon_{i}}{\sum_{j=1}^{W}p_{j(k+1)}^{q}}$

**Table 2 entropy-25-01400-t002:** Parameters *q*, β and appropriateness of fitting of the Boltzmann plot of population probability function $p_{i}$, with the constant parameter α=−1 and index of κ-distribution calculated from *q*. NDF denotes number of degree of freedom.

$T_{e}$ [eV]	*q*	β	κ	$\chi^{2}$ (NDF=39)
1	0.902	0.344	−11.2	$3.76\times 10^{-4}$
2	0.891	0.386	−10.1	$5.55\times 10^{-4}$
3	0.883	0.413	−9.56	$4.60\times 10^{-4}$
4	0.879	0.428	−9.26	$3.00\times 10^{-4}$
5	0.876	0.438	−9.06	$1.94\times 10^{-4}$
6	0.875	0.442	−8.99	$1.47\times 10^{-4}$
7	0.872	0.450	−8.83	$1.68\times 10^{-4}$
8	0.871	0.455	−8.75	$3.85\times 10^{-4}$
9	0.870	0.459	−8.69	$4.28\times 10^{-4}$
10	0.869	0.462	−8.64	$4.87\times 10^{-4}$

## Data Availability

The data that support the findings of this study are available from the corresponding author upon reasonable request.

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
