# Peer review of "Reconsideration of Temperature Determined by the Excited-State Population Distribution of Hydrogen Atoms Based on Tsallis Entropy and Its Statistics in Hydrogen Plasma in Non-Equilibrium State"

_entropy, 2023, doi:10.3390/e25101400_

Round 1
Reviewer 1 Report
In this paper the authors analyze the temperature of a non equilibrium plasma using the Tsallis entropy. They first calculate the distribution of energies using the collisional radiative (CR) model, then they analyze the resulting distributions in terms of the Tsallis entropy and compare to the standard Gibbs entropy.
I am confused as to the true significance of this work. The authors treat "temperature" as a universal property of equilibrium and non-equilibrium systems. We understand the meaning of temperature in equilibrium: temperature is given by Eq 3 in the paper and is the same temperature that appears in all equations of thermodynamics, from the ideal gas law, to the Clausius-Clapeyron equation etc. What is "temperature" in a non equilibrium state? Is it the experimental value measured by a thermometer? Does it satisfy the same fundamental equations of equilibrium thermodynamics? For example, the derivative of Gibbs's entropy with respect to volume is pressure (divided by kT), and with respect to the number of species it is chemical potential (again divided by kT). Do any of these relationships hold under Tsallis statistics? If not, what is the Tsallis temperature other than a fitted parameter?
There is another inconsistency that is not addressed in the paper. The CR model needs the temperature of the gas (Tg) and the temperature of the electrons (Te). These temperatures, both Gibbs equilibrium temperatures, are based on the assumption that the two populations, gas and electrons, equilibrate among themselves with little interaction between the two. The fact that the electron distribution that arises from the assumptions of equilibrium deviates from the Boltzmann distribution is a reflection of the approximation introduced by this assumption. In this sense the Tsallis analysis does nothing more than provide a fit for the approximations in the CR calculation.
The paper should be edited to improve English. A few examples are given here:
p2 "which does not obey Maxwell-Boltzmann distribution" (->"the" Maxwell-Boltzmann...)
p4. "Theoretical Backgrounds" (->Background)
Eq 19 is not attached to a sentence.
Very long sentence following Eq 20
Reviewer 2 Report
In this manuscript, the authors introduced Tsallis statistics to ascertain the temperature in relation to the excited-state populations within a non-equilibrium state hydrogen plasma. In fact, determining the temperature becomes intricate unless we approximate the energy-distribution function using a Maxwell-Boltzmann distribution. The authors employ a self-consistent iterative scheme involving the calculation of the q-Gaussian distribution through the application of the Hartree-Fock method algorithm. The problem addressed in this manuscript is relevant and, in my opinion, the authors provide some interesting insight into the subject. However, a number of points have to be clearly addressed before publication can be considered.
- First, one should mention that many plasma environments are in a nonequilibrium state. Attributing a single temperature to a nonequilibrium system in quite elusive since, no matter how defined, the temperature is not constant through the whole system. One way around is to describe the nonequilibrium system through two parameters: a (nonequilibrium) temperature and an index measuring how far the system is from equilibrium. Tsallis statistics used in this paper are one of such possibilities. Other possibilities are given by the concept of superstatistics (which include Tsallis statistics) and kappa-distributions.
- The concept of superstatistics is particularly relevant here because it contains Tsallis statistics as a special case. It also explains the emergence of Tsallis statistics in plasmas and other systems (see e.g., [Phys. Rev. E 100, 023205 (2019)]; [Phys. Rev. Research 2, 023721 (2020)]; [Phys. Rev. E 91, 012133 (2015); [Physica A 505, 864 (2018)]). Such analysis show part of what has been claimed in this manuscript on the different definitions of temperature. They also show how elusive it may be to describe a nonequilibrium system using a single temperature. Hence, the conclusions driven in this paper should be a bit moderated and put into perspective with the aforementioned studies.
- One last point: when the authors say "The meaning of q-parameter is the degree of the sedimentation effect that is restricted by the previously sedimented particles." One should mention that there are other interpretations as for example q arising from temperature fluctuations, long-range interactions, or memory-effects.
Reviewer 3 Report
Please find the report in the attached file.

No additional comments.
Round 2
Reviewer 1 Report
The revised manuscript has addressed my concerns.
Author Response
We are really happy to hear that the revised manuscript had addressed reviewer's concerns. We would like to express our deep gratitude to the reviewer for carefully refereeing the revised manuscript and confirming its contents.
Reviewer 2 Report
The authors made substantial changes in the manuscripts and have taken my recommendations with care. I am happy to recommend this paper in the present form.
Author Response

(The authors gave the same response as above.)

Reviewer 3 Report
I thank the authors for the answers to the queries raised in a previous report. I think now the manuscript has been improved significantly. The paper may be accepted for publication once the following comments are considered.
1. Table 2: Some comments from the authors about the small chi square values (~10^{-4}) will be welcome.
2. Lines 138-139: The q-Gaussian distribution (say f_q(x)) should contain a square of the argument x, as shown in, e.g. eq. 21 of Ref. 27 that the authors cite. So, I still do not think that the q-exponential distribution should be called a q-Gaussian.
No additional comments
